# TextRegion: Text-Aligned Region Tokens from Frozen Image-Text Models

**Yao Xiao**      **Qiqian Fu**      **Heyi Tao**      **Yuqun Wu**      **Zhen Zhu**      **Derek Hoiem**
*Siebel School of Computing and Data Science*
*University of Illinois at Urbana-Champaign*
`{yaox11, qiqianf2, heyitao2, yuqunwu2, zhenzhu4, dhoiem}@illinois.edu`

**Reviewed on OpenReview:** https://openreview.net/forum?id=KZLmkL62M4

## Abstract

Image-text models excel at image-level tasks but struggle with detailed visual understanding. While these models provide strong visual-language alignment, segmentation models like SAM2 offer precise spatial boundaries for objects. To this end, we propose `TextRegion`, a simple, effective, and training-free framework that combines the strengths of image-text models and SAM2 to generate powerful *text-aligned region tokens*. These tokens enable detailed visual understanding while preserving open-vocabulary capabilities. They can be directly applied to various downstream tasks, including open-world semantic segmentation, referring expression comprehension, and grounding. We conduct extensive evaluations and consistently achieve superior or competitive performance compared to state-of-the-art training-free methods. Additionally, our framework is compatible with many image-text models, making it highly practical and easily extensible as stronger models emerge. Code is available at: **https://github.com/avaxiao/TextRegion**.

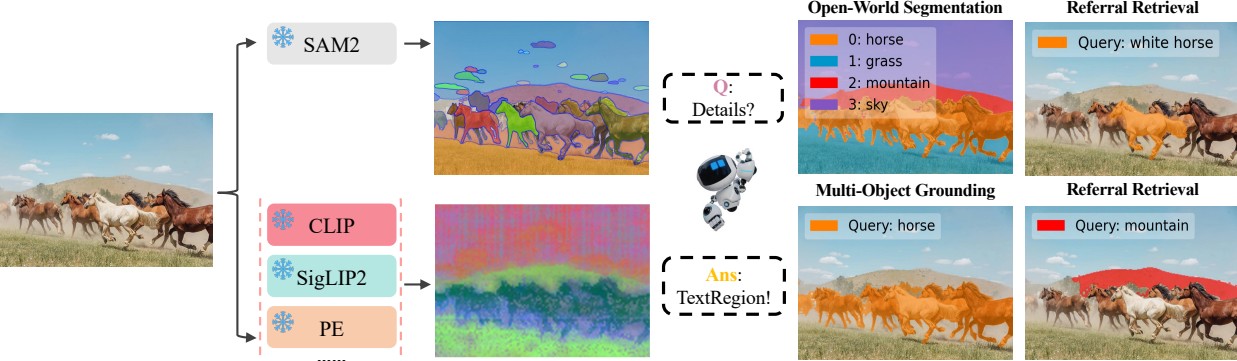

Figure 1: Using feature maps from frozen image-text models and segment masks from SAM2, we generate text-aligned region tokens that can be directly applied to various downstream tasks.

## 1 Introduction

Contrastive image-text models, such as CLIP (Radford et al., 2021), enable open vocabulary retrieval and classification for images. However, these models typically operate at the global image level. What if we are interested in classifying, localizing, or retrieving specific regions given natural language text? The ability to understand precisely at the region level remains critical for many practical vision-language tasks, including open-vocabulary semantic segmentation, visual grounding, and fine-grained retrieval.

To achieve such detailed understanding, existing methods typically fall into two categories. One group of approaches, represented by Grounding DINO (Liu et al., 2024b) and DenseCLIP (Rao et al., 2022), explicitly

trains models using detailed annotations (e.g., bounding boxes or pixel-level masks). While effective, these training-based approaches struggle to match the broad concept coverage provided by large-scale, contrastively trained image-text models, making their lack of open-vocabulary object recognition a significant limitation for real-world applications. In contrast, methods in the second category, such as MaskCLIP (Zhou et al., 2022), explore how to use existing components of the network to directly empower image-level CLIP for object-text alignment, without applying any training. Those works show that although image-text models are trained primarily to understand images at a global level, they implicitly acquire some ability for fine-grained visual–textual matching. However, patch-level embeddings are insufficient for retrieval or grounding tasks (Shlapentokh-Rothman et al., 2024), as they often lack spatial coherence and clear boundary alignment.

Given that modern segmentation models such as SAM (Kirillov et al., 2023) provide precise spatial boundaries, while image-text models like CLIP (Radford et al., 2021) capture what objects are present in images, an important question arises: *can we combine the strengths of these two families of foundation models to achieve detailed region-level understanding?* Moreover, as most prior work on detailed understanding has concentrated on CLIP, it is still uncertain whether alternative image-text encoders — such as SigLIP (Zhai et al., 2023) can effectively address this task.

In this work, we show that when supplied with accurate object masks to resolve spatial boundaries, image–text models can achieve strong zero-shot performance on detailed understanding tasks. Our `TextRegion` approach provides region-text alignment using the attention mechanism to aggregate patch features within each SAM segmented region, similar to how the image-level `CLS` token is computed for image-text alignment. We also investigate the use of multi-resolution features and propose a method to mitigate the effect of "global" tokens that are prevalent in large models. Together, TextRegion enables CLIP (Radford et al., 2021), SigLIP2 (Tschannen et al., 2025), and Perception Encoder (Bolya et al., 2025) models to generate region-level tokens that are comparable to text embeddings, often outperforming more complex methods that require additional models or retraining.

Our main contribution is a **simple, general, effective, and training-free approach to create text-compatible region tokens**, enabling powerful zero-shot region-level understanding with existing image-text models. Our approach has several benefits:

- *Excellent zero-shot performance*: Achieves impressive zero-shot results on tasks including open-world semantic segmentation, referring expression comprehension, and grounding.

- *Broad architectural compatibility*: Generalizes across diverse image-text models, with practical guidelines to ensure consistent region-text alignment.

- *Plug-and-play implementation*: Requires only a single attention layer modification and segmentation masks, making `TextRegion` immediately usable without any training.

## 2 Related Work

**Region-Level Representation.** Although patch-based representations dominate current vision research following the advent of vision transformers (Vaswani et al., 2017), region-level representations offer greater semantic richness and sparsity, often leading to superior performance in tasks such as segmentation, retrieval, and classification (Shlapentokh-Rothman et al., 2024). Recognizing their suitability for spatially detailed tasks, recent efforts have used region tokens for vision tasks (Cheng et al., 2024; Shlapentokh-Rothman et al., 2024; Pan et al., 2024; Lee et al., 2024; Sun et al., 2024). However, existing methods typically involve task-specific training. Training-free region-based methods primarily targeted tasks that do not require language understanding, such as image retrieval (Korfhage et al., 2024; Sidhu et al., 2024). To the best of our knowledge, ours is the first study to explore generating text-aligned region tokens without training.

**Open-Vocabulary Segmentation.** Recent advances in open-world segmentation like MaskCLIP (Zhou et al., 2022), SCLIP (Wang et al., 2025), ClearCLIP (Wang et al., 2025) and CLIPtrase (Lan et al., 2024a) use image-text models to achieve zero-shot segmentation by aligning patch embeddings with textual descriptions. ProxyCLIP (Lan et al., 2024b) enhances this by incorporating semantic-rich features from models like

DINO (Caron et al., 2021) and DINOv2 (Oquab et al., 2023). The current SoTA, Trident (Shi et al., 2024) further integrates segmentation architectures like SAM (Kirillov et al., 2023) and SAM2 (Ravi et al., 2024). However, combining all three large models makes their framework complex. Unlike these patch-level methods, our approach aggregates patch embeddings into region-level features, effectively transforming the dense segmentation task into a sparse region classification problem.

**Referring Expression Comprehension.** This task involves identifying image regions described by textual queries. Recent zero-shot methods (Subramanian et al., 2022; Yao et al., 2024; Yang et al., 2023; Shtedritski et al., 2023) combine proposal boxes from MAttNet (Yu et al., 2018) with image-text models for image-caption matching. Evaluating spatial relation understanding is another important aspect of this task, leading some studies to develop methods that enhance spatial reasoning in image-text models (Subramanian et al., 2022). However, our primary goal is to assess the visual-language alignment capability of our region tokens; thus, we focus exclusively on semantic retrieval without employing techniques to recover spatial relations.

**Image-Text Models.** Contrastive image-text embedding models, such as CLIP (Radford et al., 2021), ALIGN (Jia et al., 2021), SigLIP (Zhai et al., 2023), SigLIP2 (Tschannen et al., 2025), and Perception Encoder (Bolya et al., 2025), align visual and textual representations through dual encoders. These models generate text-aligned image embeddings, facilitating effective zero-shot performance across classification, text-to-image, and image-to-text retrieval. In this work, we investigate how their image-level embeddings can be transformed into region-level tokens, thus extending their applicability from global image understanding to precise region recognition.

## 3 Method

`TextRegion` extends image-text models to support region-level tasks like segmentation, localization, and retrieval. We first revisit how CLIP's class token aggregates spatial information (Sec. 3.1), noting its role in summarizing patch features. To emulate this behavior for regions, we inject spatial constraints into the transformer's cross-attention, producing text-aligned region tokens (Sec. 3.2). Finally, we introduce two useful tricks to refine region tokens (Sec. 3.3).

### 3.1 Background and Notation

Contrastive image-text models, such as CLIP (Radford et al., 2021), learn to produce an image-level token `[CLS]` that is comparable to an embedding of text that captions the image.

The model divides the input image into $N$ patches and uses a convolutional layer to map them to patch embeddings, $[\mathbf{x}_1; \mathbf{x}_2; \ldots; \mathbf{x}_N] \in \mathbb{R}^{N \times d}$, where $d$ is the embedding dimension. After adding positional embeddings, a learnable class token $\mathbf{x}_{\mathrm{CLS}} \in \mathbb{R}^d$ is prepended to the sequence, resulting in $\mathbf{X} = [\mathbf{x}_{\mathrm{CLS}}; \mathbf{x}_1; \mathbf{x}_2; \ldots; \mathbf{x}_N] \in \mathbb{R}^{(N+1) \times d}$. This sequence is processed through $L$ transformer layers. At each layer $l \in [1, L]$, the class token $\mathbf{x}_{\mathrm{CLS}}$ progressively aggregates global context via self-attention. The attention projections are computed as, $\mathbf{Q} = \mathbf{X}\mathbf{W}_q$, $\mathbf{K} = \mathbf{X}\mathbf{W}_k$, $\mathbf{V} = \mathbf{X}\mathbf{W}_v$ where $\mathbf{W}_q, \mathbf{W}_k, \mathbf{W}_v \in \mathbb{R}^{d \times d}$ denote the learned projection matrices for queries, keys, and values. The attention weights are represented by $\boldsymbol{\alpha} \in \mathbb{R}^{(N+1) \times (N+1)}$. The attention output of the class token, denoted $\mathbf{y}_{\mathrm{CLS}}$, is computed as a weighted sum over all value vectors $v$, where the weights $\alpha_{\mathrm{cls},i}$ correspond to the attention scores between $\mathbf{x}_{\mathrm{CLS}}$ and each patch token $\mathbf{x}_i$.

$$\boldsymbol{\alpha} = \mathrm{Softmax}\left(\frac{\mathbf{Q}(\mathbf{K})^\top}{\sqrt{d}}\right), \quad \mathbf{y}_{\mathrm{CLS}} = \alpha_{\mathrm{cls,cls}} v_{\mathrm{cls}} + \sum_{i=1}^{N} \alpha_{\mathrm{cls},i} v_i = \boldsymbol{\alpha}_{\mathrm{cls}} \mathbf{V}, \quad \mathbf{y}_{\mathrm{CLS}} \in \mathbb{R}^d \tag{1}$$

$\mathbf{y}_{\mathrm{CLS}}$ is passed through an MLP projection layer and added to its original feature to form the block output. The final `[CLS]` is trained to match with the text embedding of that assigned to the image.

In summary, the final-layer attention output $\mathbf{y}_{\mathrm{CLS}}$ is a weighted sum over the value vectors of all patch tokens, with attention weights $\boldsymbol{\alpha}_{\mathrm{cls}}$ determining each patch's contribution to the image `[CLS]` token.

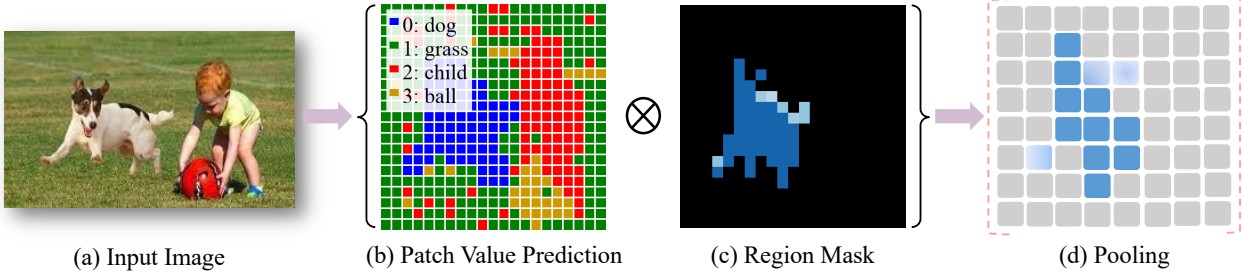

(a) Input Image      (b) Patch Value Prediction      (c) Region Mask      (d) Pooling

Figure 2: **Patch Value and Mask-based Attention Pooling**: (b) shows the segment results based on the patch value, indicating that the patch values are aligned with visual-language semantics, but could be noisy. (c) is the resized mask for a specific region, which restricts the aggregation to patches within that region. (d) demonstrates that by attending only to region-related patches, we can obtain a text-aligned region token, effectively mitigating the influence of imprecise patch values.

## 3.2 TextRegion Approach

**Key Insight.** Echoing MaskCLIP (Zhou et al., 2022), Eq. 1 suggest that the projected values from the value projection layer in the final block are quite rich in linguistic semantics, whereas the attention weights $\boldsymbol{\alpha}_{\mathrm{cls}}$ is trained to capture global information rather than local information. To enhance the region-text alignment, one key is how to organize the values. To this end, we impose *region-specific attention constraints*, restricting the [CLS] token to attend only to patches within a designated region of interest. We present the details of *region-specific attention constraints* in Fig. 2. Basically, this involves ignoring patches that are unrelated to the target region. Since most patch values within a region share the same segmentation prediction, simply pooling these patch values to form a region token can enhance the robustness of the visual tokens.

We explain how to get region tokens in the following paragraphs, with an overview shown in Fig. 3.

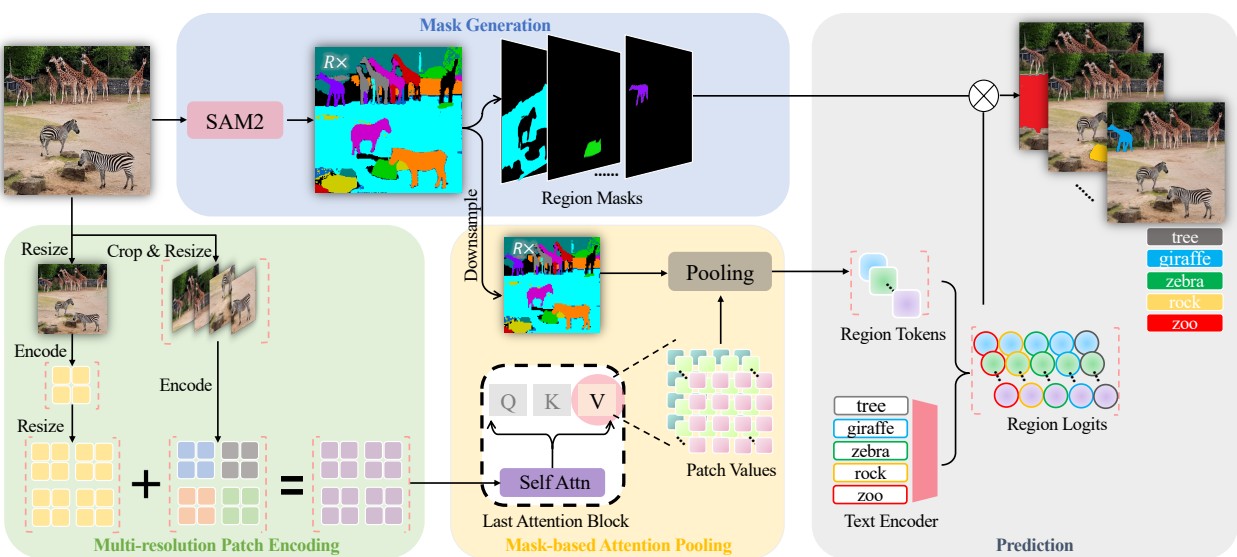

Figure 3: **TextRegion Framework**. Mask Generation: We generate $R$ soft masks using SAM2, with values ranging from 0 to 1, where each mask corresponds to a distinct region in the input image. Patch Encoding: The image is encoded to obtain a multi-resolution feature map, which is fed into the final attention block of the frozen image-text models. See Sec. 3.3 for details. Mask-based Attention Pooling: As illustrated in Fig. 2, we perform pooling based on the $R$ bilinearly downsampled masks. Prediction: Using the pooled text-aligned region tokens, we support both zero-shot region-sparse classification and dense prediction.

**Mask Generation.** For an input image of size $H \times W$, we apply SAM2 (Ravi et al., 2024) to segment it into $R$ regions. This yields a set of soft masks $\mathcal{M} = [\mathbf{M}_1; \mathbf{M}_2; \ldots; \mathbf{M}_R] \in \mathbb{R}^{R \times H \times W}$, where each mask $\mathbf{M}_r$ contains per-pixel logits $M_{h,w} \in [0,1]$. A higher value of $M_{h,w}$ indicates that pixel $(h,w)$ is more likely to belong to region $r$. We downsample the high-resolution soft mask $\mathbf{M}_r$ to the feature map resolution of the vision encoder using bilinear interpolation, obtaining $\mathbf{m_r} \in \mathbb{R}^N$. $N$ means the number of patch tokens. Each element $m_{r,i} \in [0,1]$ represents the relevance of patch $i$ to region $r$.

**Mask-based Attention Pooling.** As shown in Fig. 2, we impose locality by modulating attention scores with segmentation masks. At the final attention layer $L$, the region-specific token $\mathbf{y}_r$ selectively aggregates features by replacing the attention scores with $m_{r,i}$. We downsample the SAM2 masks to compute region tokens rather than upsample the patch feature map because 1) computing region tokens at the pixel level is resource-heavy and less robust; 2) upsampling can introduce extra noise which is problematic in the training-free setting. Additionally, we adopt soft masks instead of hard masks to better align with the attention mechanism, allowing high-confidence patches to contribute more to the aggregated tokens. The formulation for computing region tokens is:

$$\mathbf{y}_r = \sum_{i=1}^{N} m_{r,i} \, v_i = \mathbf{m_r}\mathbf{V}, \quad \mathbf{y}_r \in \mathbb{R}^d \tag{2}$$

Here, $v_i$ is the attention value feature of patch $i$ at the final layer $L$. This formulation suppresses irrelevant patches ($m_{r,i} == 0$) and aggregates features only from region-relevant ones ($m_{r,i} > 0$). The resulting region token $\mathbf{y_r}$ acts as a region-aware analogue of the global `[CLS]` token, while maintaining compatibility with the original text embedding space.

Some models, such as SigLIP (Zhai et al., 2023), compute a delegate `[CLS]` token using a final attention pooling block. Our approach also applies to these models with slight implementation differences, which we detail in Appendix Sec A.3.

**Prediction.** With the global image zero-shot classification ability maintained, `TextRegion` supports both zero-shot region classification and pixel-level prediction. For region classification, we encode candidate text labels into embeddings $\mathbf{e}_{\text{text}}$ using the pretrained text encoder. Region classification logits are computed by cosine similarity between the region token $\mathbf{y_r}$ and each text embedding, weighted by temperature $\gamma$ ($= 100$ in CLIP):

$$\mathbf{e}_{\text{text}} \in \mathbb{R}^{C \times d}, \quad \text{logits}_r = \gamma \cdot \cos(\mathbf{y}_r, \mathbf{e}_{\text{text}}), \quad \text{logits}_r \in \mathbb{R}^C, \tag{3}$$

where $C$ is the number of candidate labels. For pixel-level prediction of the target region, we restore spatial details by broadcasting the region logits back to the original image resolution. In detail, we broadcast $\text{logits}_r \in \mathbb{R}^C$ to $\mathbb{R}^{C \times H \times W}$ and then perform element-wise multiplication with the high-resolution soft SAM2 mask $\mathbf{M_r} \in \mathbb{R}^{H \times W}$:

$$\text{pred}_r^{\text{dense}} = \text{Broadcast}\left(\text{logits}_r\right) \odot \mathbf{M}_r, \quad \text{pred}_r^{\text{dense}} \in \mathbb{R}^{C \times H \times W} \tag{4}$$

This operation propagates sparse region-level logits onto their corresponding masks, yielding dense pixel-level logits for each region. To obtain the final dense prediction for the entire image, we simply sum the dense logits across all regions to produce a unified dense logit map. Note that we retain the use of soft masks $\mathbf{M}_r \in [0,1]$ during prediction, as higher mask values indicate a higher likelihood that a pixel belongs to the region. Accordingly, the logits of these pixels should more closely match the computed region logits.

### 3.3 Region Token Refinement

**Reducing global token interference**. As illustrated in Fig. 4, we find that the large model often successfully segments more challenging images but fails on simpler cases. We attribute this to the presence of *global patches.* Shao et al. (2024); Darcet et al. (2023) and Yang et al. (2025a) observe that certain tokens in vision transformers capture global image semantics rather than localized patch-level details. When aggregated via mask-based attention, such patches bias region tokens toward global text semantics, degrading localization performance. We introduce region consistency validation to filter out global patches. For each patch $i$ within

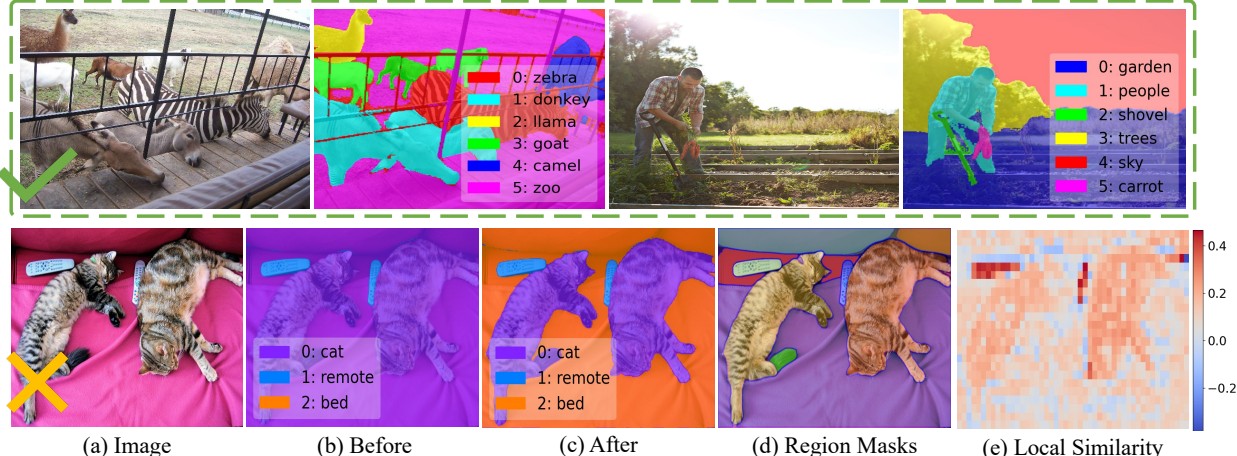

Figure 4: **Global Patches**. The first row shows segmentation examples for complex images. Despite the difficulty, the model produces correct results. In contrast, the second row presents an easier case where the model fails to segment properly. (b) and (c) show the segmentation results before and after removing global patches, respectively. (d) presents the region masks generated by SAM2, which are used to compute the local similarity defined in Eq. 5. (e) visualizes the local similarity of patches, where lower similarity indicates a higher likelihood of being a global patch. In this case, the model incorrectly classifies the bed as a cat due to the presence of many global patches in the bed area.

region $r$ (as defined by the mask $\mathbf{M}_r$), we compute both intra-region and inter-region similarity scores:

$$s_{\text{in},i} = \frac{1}{|\mathbf{P_r}|} \sum_{j \in \mathbf{P_r}} \cos\left(\mathbf{x}_i, \mathbf{x}_j\right), \quad s_{\text{out},i} = \frac{1}{|\mathbf{P}_{\neg r}|} \sum_{k \in \mathbf{P}_{\neg r}} \cos\left(\mathbf{x}_i, \mathbf{x}_k\right), \quad S_{\text{local}} = s_{\text{in},i} - s_{\text{out},i} \tag{5}$$

Here, $\mathbf{P_r}$ and $\mathbf{P}_{\neg r}$ represent the sets of patches inside and outside region $r$, respectively. A patch is identified as global if its local similarity $S_{\text{local}} = s_{\text{in},i} - s_{\text{out},i} < \tau$, where $\tau$ is a predefined threshold. Such patches are excluded from region token aggregation, ensuring that region-specific attention remains focused on locally relevant patch tokens. We apply a fixed threshold $\tau = 0.07$ across all images.

**Multi-resolution patch encoding.** Standard image-text models process fixed-resolution inputs, producing low-dimensional value feature maps that poorly capture small regions, often resulting in ineffective region masks (near-zero activations). Inspired by LLaVA's AnyResolution (Liu et al., 2024a) strategy, we propose: (1) split the original image into non-overlapping crops, resize and encode each to obtain localized features, and then concatenate them into a high-resolution feature map $V_{\text{high}}$; (2) concurrently, encode the full image to retain global context but low-resolution feature $V_{\text{low}}$. The final feature $V_{\text{final}} = V_{\text{high}} + \text{upsample}\left(V_{\text{low}}\right)$, which integrates local details with global semantics context, improving representation of both small and large regions.

## 4 Experiments

We show the strong region classification capabilities of `TextRegion` in Sec. 4.1, where our text-aligned region tokens outperform more complex methods that rely on additional models or retraining. In Sec. 4.2, we show that `TextRegion` also effectively handles instance queries, i.e., referring expression comprehension. Moreover, our framework seamlessly integrates with other image-text models, such as SigLIP2 and Perception Encoder, to produce meaningful text-aligned region tokens. Going beyond standard referring expression comprehension, which retrieves only a single region, we further demonstrate in Sec. 4.3 that our pipeline supports multiple object grounding. For all experiments, we use SAM2 (Ravi et al., 2024) with the Hiera-Large backbone to generate region masks. Additional hyperparameters are detailed in Appendix Sec A.2.

| Method | Additional Model | | With background | | | Without background | | | | | Avg |
|---|---|---|---|---|---|---|---|---|---|---|---|
| | DINO | SAM | VOC21 | Context60 | Object | VOC20 | Context59 | Stuff | City | ADE | |
| *Train CLIP ViT-B/16* | | | | | | | | | | | |
| TTD (Jo et al., 2024) | | | 61.1 | **37.4** | **37.4** | - | - | 23.7 | 27.0 | 17.0 | - |
| CLIP-DINOiser (Wysoczańska et al., 2024) | ✓ | | **62.1** | 32.4 | 34.8 | **80.9** | 35.9 | 24.6 | **31.7** | **20.0** | 40.3 |
| SAM-CLIP (Wang et al., 2024b) | | ✓ | 60.6 | 29.2 | - | - | - | **31.5** | - | 17.1 | - |
| *Training-Free, CLIP ViT-B/16* | | | | | | | | | | | |
| MaskCLIP (Dong et al., 2023) | | | 43.4 | 23.3 | 20.6 | 74.9 | 26.4 | 16.7 | 24.9 | 11.9 | 30.3 |
| SCLIP (Wang et al., 2024a) | | | 59.1 | 30.4 | 30.5 | 80.4 | 34.2 | 22.4 | 32.2 | 16.1 | 38.2 |
| ResCLIP (Yang et al., 2025b) | | | - | - | - | **86.0** | 36.8 | 24.7 | 35.9 | 18.0 | - |
| ProxyCLIP (Lan et al., 2024b) | ✓ | | 61.3 | 35.3 | 37.5 | 80.3 | 39.1 | 26.5 | 38.1 | 20.2 | 42.3 |
| LaVG (Kang & Cho, 2024) | ✓ | | 62.1 | 31.6 | 34.2 | 82.5 | 34.7 | 23.2 | 25.0 | 15.8 | 38.6 |
| LPOSS+ (Stojnić et al., 2025) | ✓ | | 62.4 | 35.4 | 34.3 | 79.3 | 38.6 | 26.5 | 37.9 | 22.3 | 42.1 |
| Trident (Shi et al., 2024) | ✓ | ✓ | 67.1 | 38.6 | 41.1 | 84.5 | 42.2 | 28.3 | **42.9** | 21.9 | 45.8 |
| CLIPtrase (Shao et al., 2024) | | ✓ | 57.1 | 32.0 | **44.2** | 82.2 | 36.4 | 24.8 | - | - | - |
| TextRegion | | ✓ | **70.9** | **39.1** | 41.1 | 84.4 | **43.2** | **28.7** | 42.8 | **22.8** | **46.6** |
| *Training-Free, OpenCLIP ViT-H/14* | | | | | | | | | | | |
| SCLIP (Wang et al., 2024a) | | | 43.8 | 23.5 | 24.6 | 67.5 | 25.6 | 16.8 | 19.5 | 11.3 | 29.1 |
| ProxyCLIP (Lan et al., 2024b) | ✓ | | 65.0 | 35.4 | 38.6 | 83.3 | 39.6 | 26.8 | 42.0 | 24.2 | 44.4 |
| Trident (Shi et al., 2024) | ✓ | ✓ | 70.8 | 40.1 | **42.2** | 88.7 | 44.3 | 28.6 | **47.6** | 26.7 | 48.6 |
| TextRegion | | ✓ | **73.1** | **41.2** | 40.6 | **89.5** | **46.1** | **31.2** | 47.0 | **27.3** | **49.5** |

Table 1: **Open-world Semantic Segmentation.** Evaluation results (mIoU) on semantic segmentation benchmarks. `TextRegion` consistently achieves superior or competitive performance across various benchmarks and backbone architectures. The best results in each setting are **bolded**.

## 4.1 Open-world Semantic Segmentation

**Dataset.** We evaluate on six widely used semantic segmentation benchmarks: PASCAL VOC 2012 (Everingham et al., 2015), PASCAL Context (Mottaghi et al., 2014), COCO-Stuff (Caesar et al., 2018), COCO-Object (Lin et al., 2014), Cityscapes (Cordts et al., 2016), ADE20K (Zhou et al., 2019). For VOC and Context, we test the settings without (VOC20, Context59) and with (Voc21, Context60) background labels. Since the image resolutions vary across datasets, we follow the design of previous works by setting different resizing scales for each dataset. Specifically, the shorter side is resized to 672 pixels for VOC20, VOC21, COCO-Object, Context59, ADE, and Context60; 896 pixels for COCO-Stuff; and 1344 pixels for Cityscapes.

**Results.** Tab. 1 shows our open-world semantic segmentation performance. The primary distinction between our method and baselines lies in the adoption of a region-level classification strategy prior to segmentation. Specifically, we first classify those region tokens, and then propagate the region logits back to the pixel-level segmentation predictions, leveraging the fact that our region tokens are inherently associated with high-resolution segmentation masks. In contrast, other methods typically conduct predictions at the patch level and subsequently rely on upscaling or additional post-processing to produce high-resolution segmentation results. Benefiting from the open-vocabulary classification capacity preserved in our region tokens and the strong segmentation capabilities of SAM2, `TextRegion` effectively simplifies the dense prediction task into an instance-level sparse prediction problem, which is both easier to solve and more robust. Despite being entirely training-free and simple, our approach consistently achieves superior or competitive performance.

## 4.2 Zero-shot Referring Expression Comprehension

**Dataset.** We evaluate on RefCOCO (Yu et al., 2016), RefCOCO+ (Yu et al., 2016) and RefCOCOg (Mao et al., 2016) datasets. All of these benchmarks come from the MS-COCO (Lin et al., 2014) dataset, paired with expressions that refer to a unique object in each image, accompanied with a bounding box. The main difference is that RefCOCO+ excludes relation-based expressions (e.g. "left of", "closer", or "bigger") and RefCOCOg has longer descriptions. The test sets of RefCOCO and RefCOCO+ are divided into "testA" and "testB," which contain only people and non-people instances, respectively.

**Setting.** Referring Expression Comprehension (ReC) focuses on identifying the most relevant object (bounding box) in an image based on a given natural language expression. A common zero-shot ReC setting

| Method | Backbone | RefCOCO | | | RefCOCO+ | | | RefCOCOg | |
|---|---|---|---|---|---|---|---|---|---|
| | | Val | TestA | TestB | Val | TestA | TestB | Val | Test |
| CPT (Yao et al., 2024) | VinVL (Zhang et al., 2021) | 32.3 | 36.1 | 30.3 | 31.9 | 35.2 | 28.8 | 36.7 | 36.5 |
| TextRegion | SigLIP2-L/16 (Tschannen et al., 2025) | 45.0 | 49.8 | 36.0 | 51.3 | 57.1 | 40.5 | **53.8** | **52.6** |
| TextRegion | PE-Core-L/14 (Bolya et al., 2025) | **47.3** | **53.7** | **37.9** | 52.6 | 59.7 | 42.2 | 52.7 | 50.7 |
| Red Circle (Shtedritski et al., 2023) | CLIP ViT-L/14@336 | 38.0 | 45.3 | 32.9 | 43.9 | 51.0 | 37.1 | 47.2 | 47.3 |
| FGVP (Yang et al., 2023) | CLIP ViT-L/14@336 | 46.1 | 53.0 | 40.4 | 50.4 | 57.5 | 42.6 | 54.5 | 54.1 |
| TextRegion | CLIP ViT-L/14@336 | **48.7** | **56.4** | **40.8** | **53.6** | **60.8** | **44.3** | **55.8** | **54.6** |

Table 2: **Zero-shot Referring Expression Comprehension.** `TextRegion` is compatible with different image-text models and consistently achieves strong zero-shot results across all benchmarks.

leverages a pretrained object detector to generate object proposals and then selects the most likely proposal corresponding to the query. The accuracy is evaluated by the percentage of instances where the selected proposal has an Intersection-over-Union (IoU) of at least 0.5 with the ground-truth bounding box. For this experiment, we do not incorporate any spatial relationship refinement or post-processing methods commonly used in ReC. Retrieval accuracy is computed using object proposals produced by MAttNet (Yu et al., 2018). All baseline results are taken from FGVP (Yang et al., 2023). For a fair comparison, we evaluate `TextRegion` with the same set of object proposals as those used by FGVP.

For our implementation, we first generate region tokens for the input image and then compute the cosine similarity between the text-aligned region tokens and the query text embedding. The region with the highest similarity is selected as the retrieved object, and its corresponding bounding box is determined. To select the most likely candidate from the detected proposals, we calculate the IoU between our predicted region bounding box and all proposal bounding boxes, selecting the proposal with the highest IoU. In cases where none of the proposals overlap with the selected region, we directly return the region's bounding box as the retrieval result.

**Results.** Tab. 2 shows the results for zero-shot ReC. Complex baselines depend on carefully designed pipelines and prompting strategies (Yao et al., 2024; Shtedritski et al., 2023), and using bounding box candidates as prompts for SAM to obtain better object masks (Yang et al., 2023). In contrast, our method directly generates region tokens without relying on candidate bounding boxes and still achieves superior performance.

### 4.3 Multiple Object Grounding

**Setting.** In Sec. 4.2 we test the referring expression comprehension performance of our method. But since that task involves referring to a single target object, we are interested in testing our method's multi-object grounding capabilities. For this test, we use the Reasoning Segmentation dataset (Lai et al., 2024) which is commonly used to assess the visual segmentation capabilities of VLMs. As some samples in the dataset contain multiple objects related to a given query, simply returning the region with the highest similarity is not effective and can overlook relevant objects.

To assess the relevance of a region to a given query, we create a negative label to get a score for the query. Specifically, for each input query, we generate a pseudo-contrastive query to compute region similarities relative to both the original and contrastive queries. Regions whose similarity to the original query exceeds their similarity to the pseudo-contrastive query are retrieved as the selected objects. The pseudo-contrastive query we use for the original reasoning segmentation query is *"Background, any other thing"*.

Since the original queries in the reasoning segmentation dataset are long and complex, we construct a simplified scenario. Specifically, we use the original LLaVA1.5-7B (Liu et al., 2024a) to generate text answers for the original queries, and then treat these answers as interpreted queries to retrieve relevant objects. This scenario is designed to evaluate the impact of language understanding ability. Further details are provided in Appendix Sec A.4. We implement baseline results for Trident (Shi et al., 2024), while the results for all other baselines are sourced from LISA (Lai et al., 2024). We use CLIP ViT-L/14@336px as the backbone for this experiment.

| Method | Short query gIoU | cIoU | Long query gIoU | cIoU | Overall gIoU | cIoU |
|---|---|---|---|---|---|---|
| *Train LLaVA1.5-7B* | | | | | | |
| LISA (Lai et al., 2024) | 47.1 | 48.5 | 49.2 | 48.9 | 48.7 | 48.8 |
| *Training-based (No VLM used)* | | | | | | |
| OVSeg (Liang et al., 2023) | 18.0 | **15.5** | **28.7** | **22.5** | **26.1** | **20.8** |
| X-Decoder (Zou et al., 2023a) | **20.4** | 11.6 | 22.2 | 17.5 | 21.7 | 16.3 |
| SEEM (Zou et al., 2023b) | 20.1 | 11.5 | 25.6 | 20.8 | 24.3 | 18.7 |
| Grounded-SAM (Liu et al., 2024b) | 17.8 | 10.8 | 22.4 | 18.6 | 21.3 | 16.4 |
| *Training-free (No VLM used)* | | | | | | |
| Trident (Shi et al., 2024) | 19.9 | 12.9 | 24.5 | 23.1 | 23.3 | 20.0 |
| TextRegion | **21.6** | **15.6** | **26.4** | **24.1** | **25.2** | **21.6** |
| *With interpreted queries from the original LLaVA1.5-7B* | | | | | | |
| OVSeg (Liang et al., 2023) | 24.2 | 18.7 | 44.6 | 37.1 | 39.7 | 31.8 |
| Trident (Shi et al., 2024) | 23.0 | 24.2 | 43.3 | 42.5 | 38.4 | 39.2 |
| TextRegion | **28.5** | **30.6** | **47.2** | **45.3** | **42.7** | **42.4** |

Table 3: **Multiple Object Grounding.** The last three rows show results using interpreted queries. `TextRegion` demonstrates significant performance gains when given LLaVA-interpreted queries, outperforming baselines by a large margin.

Original: In case of a fire, it is important to have access to fire safety equipment. What object in the picture is specifically designed to store and release fire extinguishing substances?

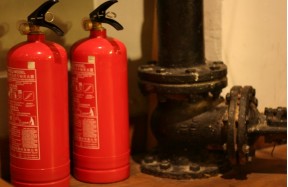

Interpreted: fire extinguisher

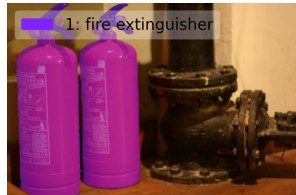

Figure 5: Top: Query image; Bottom: Grounding results. The title shows the original / interpreted query.

**Metrics.** There are two metrics for this task: gIoU and cIoU. gIoU is defined as the average of per-image Intersection-over-Unions (IoUs), while cIoU is computed as the cumulative intersection divided by the cumulative union. Since cIoU is heavily biased toward large-area objects and exhibits high variability, gIoU is usually preferred (Lai et al., 2024).

**Results.** As shown in Tab. 3, `TextRegion` achieves performance comparable to training-based methods. The performance improves significantly when interpreted queries are used, especially for long queries. This suggests that the main bottleneck for `TextRegion` in reasoning segmentation stems from the limited language understanding of the image-text models' text encoder, rather than from its object retrieval capability.

## 5 Ablation and Discussion

**Global Patch, Multi-resolution and CLIP variants.** We validate the effectiveness of the proposed global patch removal and multi-resolution patch feature strategies in Tab. 4. Overall, these strategies benefit a range of image-text models, from small to large backbones. When using both techniques, the performance is generally the best compared to using alone. An interesting observation is that the performance of SigLIP2-

| Global | Multi | COCO Stuff Perception B/16 | L/14 | SigLIP2 B/16 | SO/16 | CLIP B/16 | H/14 | AED20K Perception B/16 | L/14 | SigLIP2 B/16 | SO/16 | CLIP B/16 | H/14 |
|---|---|---|---|---|---|---|---|---|---|---|---|---|---|
| | | 24.0 | 23.8 | 23.6 | 22.4 | 26.9 | 28.7 | 20.5 | 23.3 | 23.3 | 23.7 | 20.6 | 24.1 |
| ✓ | | 25.1 | 24.2 | 24.7 | **23.7** | 27.0 | 28.6 | 21.1 | 23.6 | 23.9 | **24.6** | 20.7 | 23.8 |
| | ✓ | 27.7 | 25.4 | 26.6 | 21.4 | **28.8** | 30.6 | 24.1 | 24.3 | 25.3 | 21.5 | 22.7 | 26.5 |
| ✓ | ✓ | **27.9** | **26.1** | **26.8** | 23.2 | 28.7 | **31.2** | **24.3** | **24.6** | **26.0** | 22.1 | **22.8** | **27.3** |

Table 4: **Ablation on Global Patch Removal and Multi-Resolution Feature for CLIP Variants.** Combining global patch removal with multi-resolution features achieves the best performance in most cases. Our default setting is marked in green.

| Feature | Stuff | ADE |
|---|---|---|
| Input | 0.9 | 1.1 |
| Output | 5.7 | 3.0 |
| Value | **28.7** | **22.8** |

Table 5: **Attn Feature.**

| Layer | Stuff | ADE |
|---|---|---|
| -5 | 0.2 | 0.4 |
| -2 | 18.2 | 11.6 |
| -1 | **28.7** | **22.8** |

Table 6: **Value Layer.**

| Interp | mIoU | s/img |
|---|---|---|
| Up | 10.2 | 0.33 |
| Down | **28.7** | **0.20** |

Table 7: **Interpolate.**

| Mask | Stuff | ADE |
|---|---|---|
| Hard | 28.5 | 22.7 |
| Soft | **28.7** | **22.8** |

Table 8: **Mask Value.**

SO400M/16 drops when using the multi-resolution patch feature, whereas all other models benefit from it. A possible explanation is that large backbones, while powerful, tend to generate more global patches (Xiao et al., 2023; Darcet et al., 2023; Shao et al., 2024). When applying the multi-resolution feature alone, this effect may amplified — simpler images contain less detailed information, causing more patches to capture global features. Our global patch removal strategy helps mitigate the negative effects, so when combined with multi-resolution patch features, it consistently delivers better or comparable performance than using multi-resolution alone.

Another interesting finding is that the global patch phenomenon is less severe in CLIP compared to the perception encoder and SigLIP2. As a result, using global patch removal alone for CLIP has a limited impact. However, when combined with multi-resolution features, it significantly boosts performance, particularly for CLIP ViT-H/14.

**Design choice for mask-based attention pooling.** We evaluate different design choices for mask-based attention pooling using CLIP ViT-B/16. Tab. 5 compares the effect of different feature used for pooling. "Input" refers to the feature map fed into the final attention block, "Output" denotes the output of that block, and "Value" means the attention values from the last attention block. Interestingly, even with a single projection, transitioning from the input to the value features significantly improves performance—from almost zero to decent results.

Tab. 6 shows the effect of pooling attention value from different layers. Results show that the value features from the second-to-last layer yields substantially better performance than directly pooling the input feature map (Tab. 5) for the last attention block.

Tab. 7 shows the results of either upscaling the feature maps or downscaling the SAM2 masks when pooling region tokens. Experiments are conducted on COCO-Stuff. The reported "Up" result use a $5\times$ upscaling of the value feature maps instead of directly matching the shape of the SAM2 masks, as the latter approach can induce much larger GPU consumption, particularly when combined with the multi-resolution strategy. "s/img" means seconds required to predict a single image, measured on one A100 GPU. The results show that downsampling offers both effectiveness and efficiency advantages. Tab. 8 shows the impact of using hard masks versus soft masks for aggregating region tokens and computing region logits.

**Sensitivity to Mask Quality.** Tab. 9 reports the impact of different mask generators. "Ground Truth Masks" means using the annotated ground-truth masks to compute region tokens, serving as the upper bound of `TextRegion`. In contrast, SLIC (Achanta et al., 2012) partitions images into superpixels based on

| Mask Generator | COCO Stuff | | | AED20K | | |
|---|---|---|---|---|---|---|
| | Perception | SigLIP2 | CLIP | Perception | SigLIP2 | CLIP |
| Ground Truth masks | 34.7 | 33.7 | 36.5 | 32.7 | 35.4 | 31.0 |
| SLIC (Achanta et al., 2012) | 21.7 | 19.9 | 21.3 | 18.4 | 18.2 | 16.1 |
| SAM (Kirillov et al., 2023) | 27.0 | 26.1 | 27.9 | 24.3 | 25.8 | 22.1 |
| SAM2 (Ravi et al., 2024) | 27.9 | 26.8 | 28.7 | 24.3 | 26.0 | 22.8 |

Table 9: **Ablation on the effect of mask generators.** "Ground Truth masks" refer to using the annotated masks to extract region tokens, which are subsequently used for prediction. All experiments are conducted with image-text model based on the ViT-B/16 backbone. The default configuration is highlighted in green.

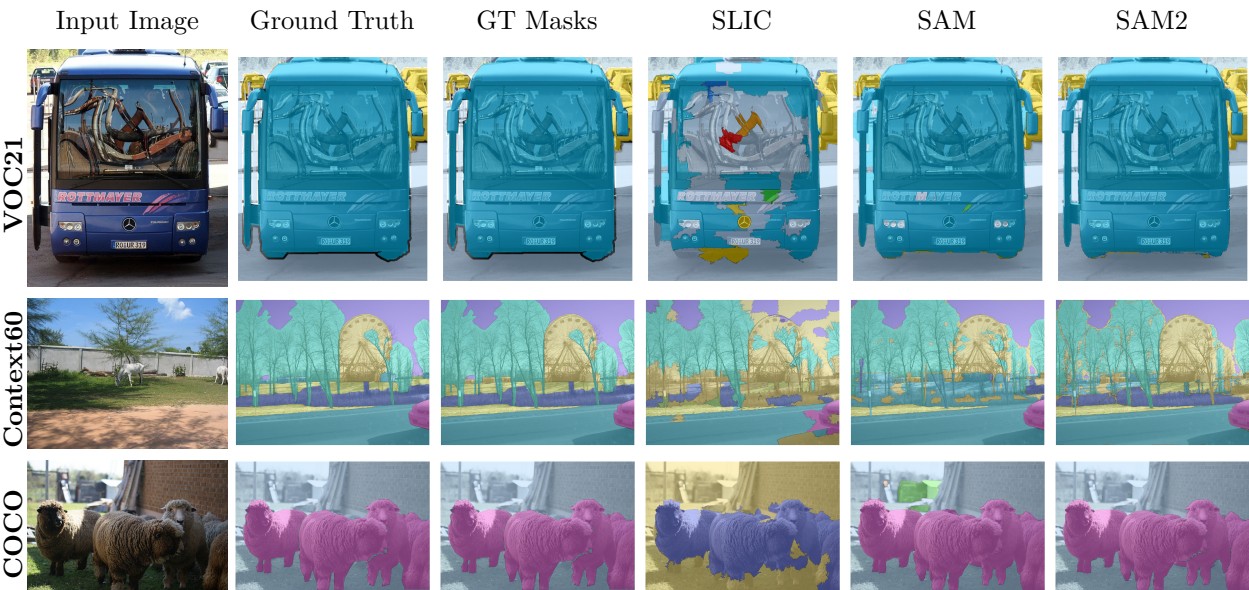

Figure 6: **Visualization comparison of different mask generators across datasets.** "GT masks" refer to using the annotated masks to extract region tokens.

color and spatial cues, resulting in lower performance than SAM and SAM2. These results demonstrate that `TextRegion` is compatible with various segmentation methods, enabling object-level predictions for flexible, user-specified mask inputs. We also provide visual comparisons of those mask generators in Fig. 6.

## 6  Conclusion

`TextRegion` is a simple, effective, and training-free approach for obtaining text-aligned region tokens by combining image-text models with SAM2. The image-text model provides visual-language alignment capabilities, while SAM2 supplies detailed object spatial boundaries. Our region tokens show strong zero-shot performance on open-world semantic segmentation, referring expression comprehension, and grounding. Our method is also highly flexible, supporting easy integration with various image-text models, all of which perform well in zero-shot settings. Moreover, `TextRegion` imposes no constraints on the masks, users can manually define custom masks to generate region tokens for any objects of interest, beyond those provided by SAM2.

**Limitations**: The quality of the region tokens depends on the accuracy of the region masks. Poor masks will produce low-quality region tokens. Additionally, the effectiveness of region tokens is limited by the original visual feature of the image-text model. If these feature lack spatial awareness or advanced language understanding, the region tokens will also be unable to capture such information.

**Acknowledgement**: This work is supported in part by ONR N00014-23-1-2383 and DARPA HR0011-23-9-0060. This work used NVIDIA GPUs at NCSA Delta through allocation CIS240059 and CIS250059 from the Advanced Cyberinfrastructure Coordination Ecosystem: Services & Support (ACCESS) program, which is supported by NSF Grants #2138259, #2138286, #2138307, #2137603, and #2138296. The views and conclusions expressed are those of the authors, and not necessarily representative of the US Government or its agencies.

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

# A  Appendix

## A.1  Summary of contents

Sec. A.2 details the hyperparameters used in our experiments. In Sec. A.3, we describe how to generate text-aligned region tokens using the frozen SigLIP2 (Tschannen et al., 2025) and Perception Encoder (Bolya et al., 2025). Sec. A.4 explains the process of generating interpreted queries for Reasoning Segmentation. Finally, Sec. A.5 includes further visual examples for open-world segmentation and referring expressions, as well as illustrative failure cases.

## A.2  Hyperparameter

For all experiments, we filter global patches using a threshold of $\tau = 0.07$. The crop size is uniformly set to 336 for all CLIP models (ViT-B/16 through ViT-H/14), while SigLIP2 and Perception Encoder use their respective default input resolutions. Region masks are generated with SAM2 (Ravi et al., 2024) Hiera-Large, using the following configuration: `pred-iou-thresh` set to 0.6, `stability-score-thresh` to 0.6, `box-nms-thresh` to 0.9, and `points-per-side` to 16. In the semantic segmentation experiments on the Cityscapes dataset, we increase `points-per-side` to 36 to due to its high resolution and the abundance of small objects. To mitigate the impact of duplicated or overlapping masks, we also merge masks with an overlap IoU greater than 0.8.

## A.3  CLIP Variants

Unlike CLIP (Radford et al., 2021), which retains the `[CLS]` token across multiple layers, SigLIP2 (Tschannen et al., 2025) and Perception Encoder (Bolya et al., 2025) introduce a learnable query token, denoted as `[q]`, to aggregate global image information within the final attention pooling block. Let $X$ denote the input to this last attention pooling block:

$$\text{CLS} = \text{Attn}\left(\underbrace{q}_{\text{query}}, \underbrace{\mathbf{XW}_k}_{\text{key}}, \underbrace{\mathbf{XW}_v}_{\text{value}}\right) \tag{6}$$

Note that the core idea of `TextRegion` is to restrict the *query*'s attention to region-relevant patches. To simplify the implementation, instead of explicitly computing the *value* and then do pooling, we reformulate region-aware pooling by directly masking out irrelevant patches during the attention computation. Given the SAM2 down-sampled region mask $\mathbf{m_r} \in \mathbb{R}^N$, the text-aligned region token $\text{CLS}_r$ is computed as:

$$\text{CLS}_r = \text{Attn}\left(q, \bar{\mathbf{X}}\mathbf{W}_k, \mathbf{XW}_v\right), \quad \text{attention mask} = (-\inf * (\mathbf{m_r} == 0)) \tag{7}$$

$\bar{\mathbf{X}}$ denotes the average of all patch features, to reduce dependence on the original attention scores and better approximate our mask-based attention pooling in CLIP, where mask values within a region are typically close to one. Experiments show that using the original $\mathbf{XW}_k$ as the key also works well. Although we have not conducted a comprehensive comparison between the two approaches, visualizations suggest the averaging method performs slightly better. Given its alignment with our CLIP implementation—which avoids relying on original attention scores—we adopt it as the default for both SigLIP2 and the Perception Encoder.

Recall that $m_r$ is obtained by bi-linearly downsampling $M_r$, where the original $M_r \in [-32, 32]$ but are clamped to $[0, 1]$ before use. As a result, each downsampled value $m_{r,i} \in [0, 1]$ represents the likelihood that patch $i$ belongs to the region $r$. When $m_{r,i} == 0$, patch $i$ is considered irrelevant to the region $r$ and should not be attended to by the region query. To enforce this constraint, we apply an *attention mask* that excludes patches with $m_{r,i} == 0$ from participating in the attention computation.

Another modification for SigLIP2 (Tschannen et al., 2025) and the Perception Encoder (Bolya et al., 2025) is that we change the final value feature to $V_{\text{final}} = V_{\text{high}} + 0.5 * \text{upsample}\left(V_{\text{low}}\right)$ for the multi-resolution strategy, as they are more sensitive to the influence of global patches.

### A.4 Interpreted Queries for Multi-Object Grounding / Reasoning Segment

To convert the original long and complex queries in the reasoning segmentation dataset into interpretable queries compatible with the CLIP text encoder, we use LLaVA1.5-7B (Liu et al., 2024a) for query preprocessing. Specifically, we input the paired image and query into LLaVA1.5-7B and obtain a concise answer using the prompt: *"Please summarize the answer in five words or fewer to define the object."* The answer is then used as the interpreted query to obtain our multi-object grounding results. See Fig. 7 and Fig. 8 for more examples of interpreted queries.

This interpreted setup allows us to evaluate visual grounding performance independently of advanced language understanding capabilities. Since the reasoning grounding performance depends on both the interpreted queries and the object grounding capability of `TextRegion`, we also present a case in Fig. 8 (example f) where LLaVA produces an incorrect interpreted query.

The template used to construct the contrastive query for the interpreted new query is: *"Background, anything but {interpreted query}"*. In the main paper, we also evaluate the grounding performance on the original complex queries. We do not apply this template (*"Background, anything but {original query}"*) to get the contrastive queries because the original query are too long, and adding them to the contrastive query would further burden the text encoder. Instead, we simply use *'Background, any other thing"* as the contrastive query for these cases.

(a) Original: Generally speaking, dogs do not have horns on their heads, only a pair of ears. What part of the dog's head in this picture looks strange?

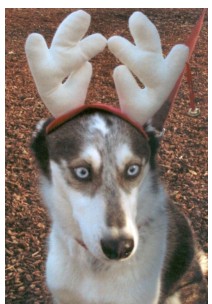 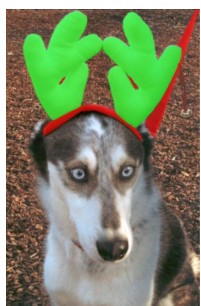 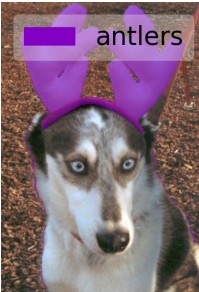

(b) Original: Driving at night can be very dangerous due to poor visibility, which can lead to accidents. What part of the car needs to be turned on when driving at night?

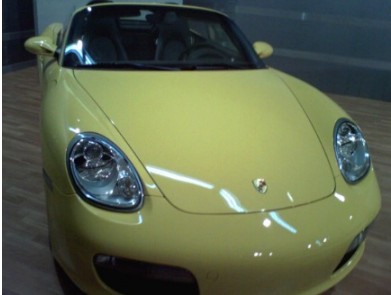 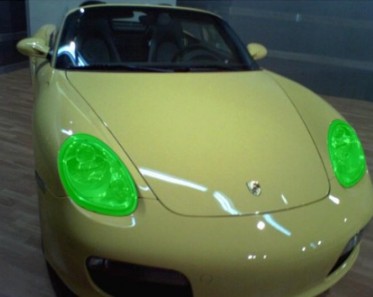 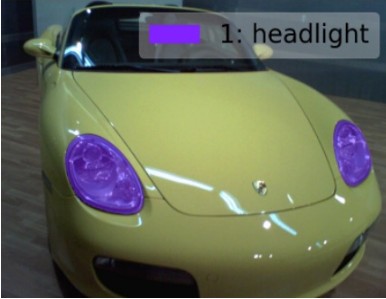

Figure 7: **Examples for Interpreted Queries (Part 1)**: The title means the original query from the reasoning segmentation dataset, while the legend presents the interpreted query generated by LLaVA 1.5. The left column displays the query image. The middle column shows the ground truth, with green highlighting the target grounding area, and red indicates areas that can be ignored. The right column presents the grounding result.

(c) Original: In a rural landscape, what objects in the picture could provide shade and shelter for animals or humans?

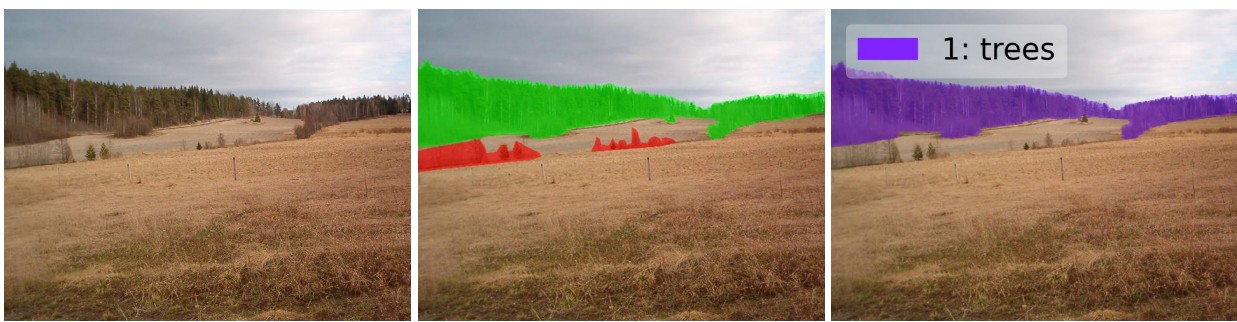

(d) Original: a car with a color that is closer to lipstick color

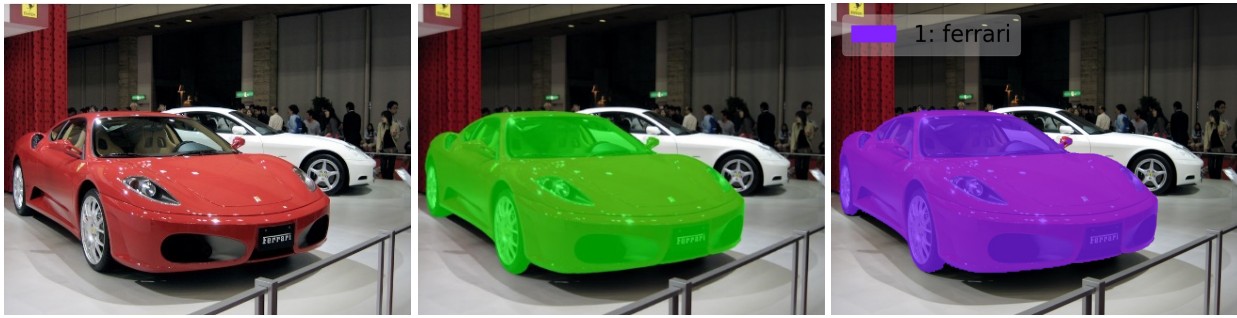

(e) Original: To keep bread fresh and protected, it is often placed in a protective covering. What item in the picture is commonly used for this purpose?

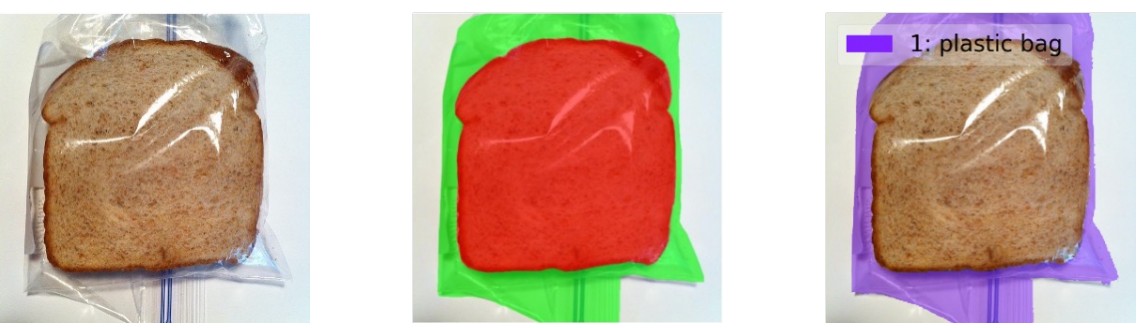

(f) Original: the shadow of the red car

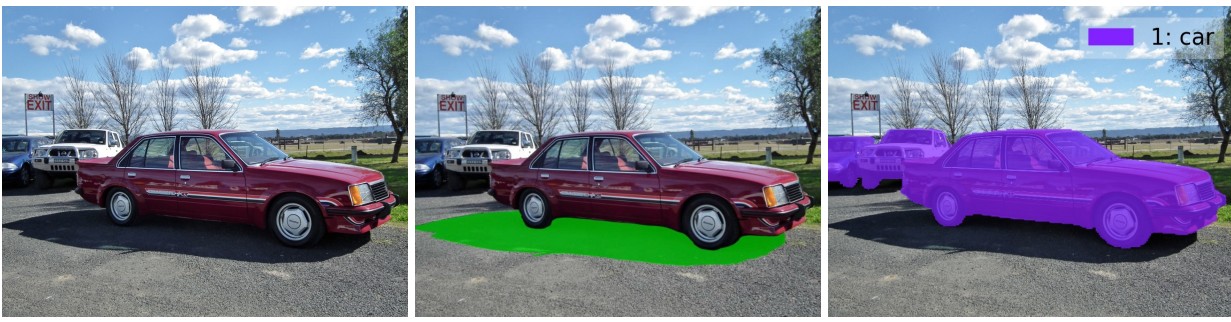

Figure 8: **Examples for Interpreted Queries (Part 2)**: The middle column shows the ground truth: red highlights areas that can be ignored, and green indicates the target grounding area. Example (f) shows a failure case where LLaVA1.5 provides an incorrect answer to the original query.

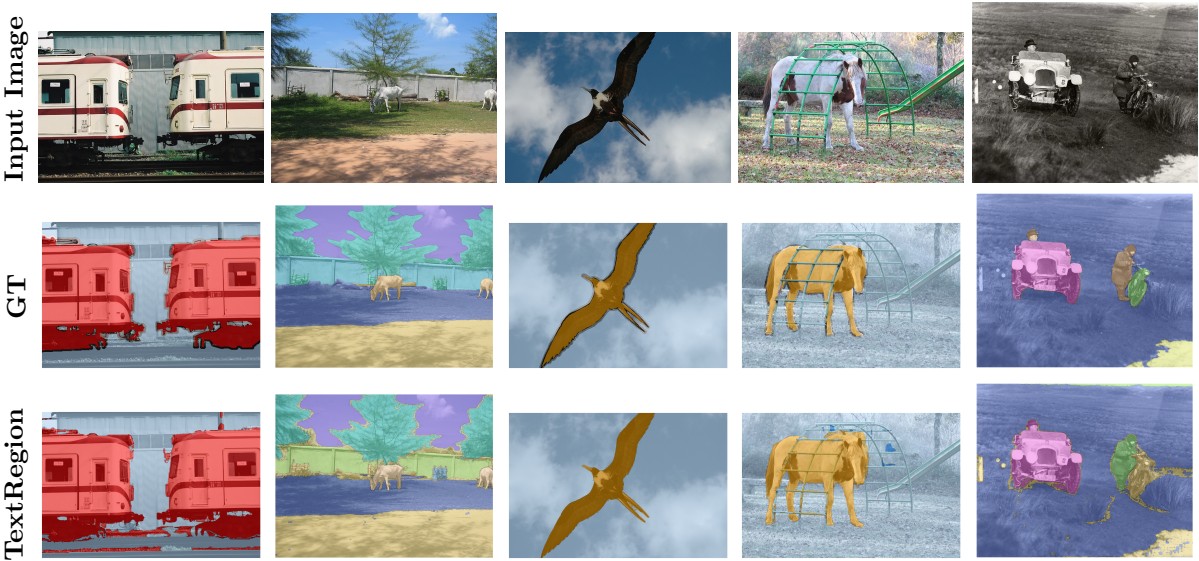

Figure 9: **Visualization for open-world segmentation.** "GT" means the ground truth, while "TextRegion" shows our predicted results.

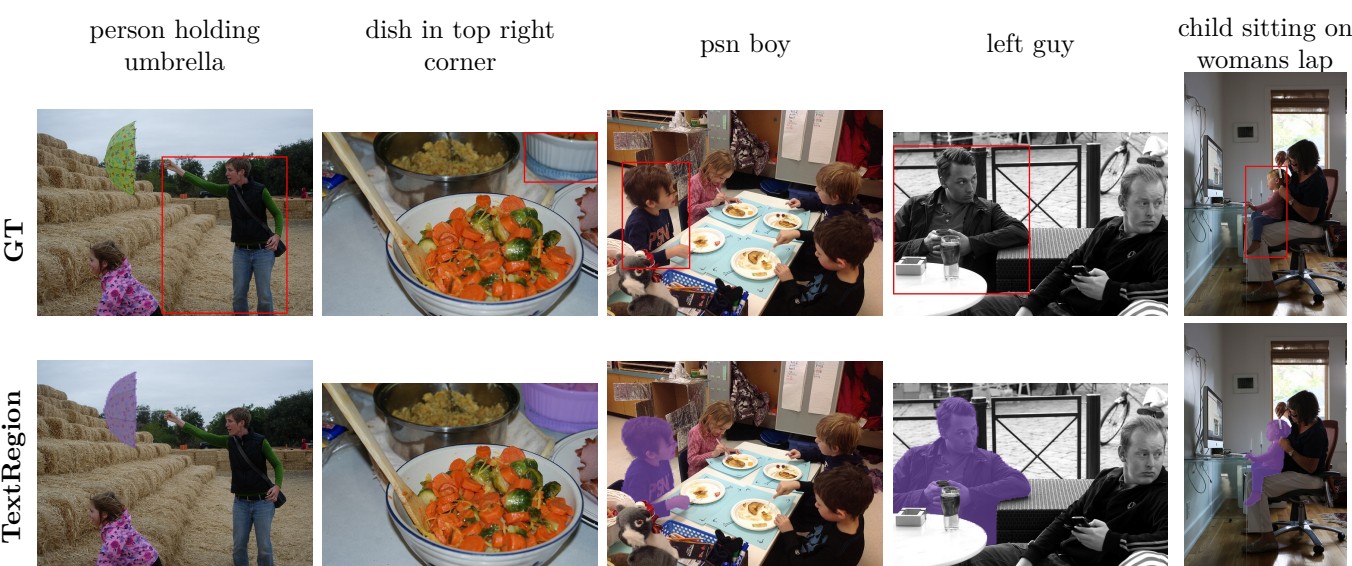

Figure 10: **Visualization for referring expressions.** The sentences above are the query expressions. The red bounding box in the "GT" row indicates the referred object.

### A.5 Visualization

Fig. 9 and Fig. 10 presents additional visualizations for open-world segmentation and referring expression tasks. From the visual results, we observe that while SAM2 can segment objects with relatively accurate boundaries, it struggles with objects exhibiting complex contours. For instance, in the train example of Fig. 9, `TextRegion` produces slight pixel-level inaccuracies along the intricate edges.

For referring expressions, the example of "person holding umbrella" shows that the text encoder may sometimes misinterpret the expression, leading the model to predict a related object rather than the specific target described.

