# OpenReview forum: "TextRegion: Text-Aligned Region Tokens from Frozen Image-Text Models"
_TMLR — Accepted by TMLR_

### Review · Reviewer_g48z · 2025-09-15

**Summary Of Contributions:**

The authors present a framework that leverages image-text models to generate region tokens for enhanced visual understanding in referring expression comprehension tasks. The proposed method substitutes the attention weights in the final layer with region-specific tokens derived from SAM2 to direct the text encoder.

Strengths:
* Training-free framework that can be readily adapted to enhance pre-trained image-text models.
* Superior performance compared to text embedding models that require additional training.

Weaknesses:
* Heavy reliance on region tokens from SAM2. Although the authors evaluate different image-text models (CLIP, SigLIP2, OpenCLIP etc.), they do not investigate the framework's robustness when using region tokens from alternative segmentation models, such as the original SAM or other less capable variants.

**Additional Comments:**

The use of imagery containing weapons in Appendix A.4 may be inappropriate for an academic paper.

**Audience:**

Yes

**Audience Explanation:**

Text-image models are of interest to a portion of the TMLR audience

**Claims And Evidence:**

Yes

**Claims Explanation:**

The authors compare their framework against both training-free and trained methods.

Insufficient empirical support: While the reviewer acknowledges that certain patches likely indeed contain global information, the authors make an unsubstantiated claim by asserting that tokens lacking patch-level local information must necessarily represent global information, without providing adequate empirical evidence to support this binary classification.

**Requested Changes:**

The authors claim certain patches contain 'global' information but provide no evidence beyond showing that patches with low local similarity (below τ = 0.07) are classified as 'global.' They should demonstrate that these patches actually encode global semantic content rather than simply being noisy or representing object boundaries.

---

> ### Author Response · Authors · 2025-10-05
>
> Thank you for your review and affirming the supported claims and interest. We had not intended to imply that the existence of "global patches" is a new claim of our paper, as the presence of these patches has been extensively explored and mitigated or exploited in previous works, including the Shao et al. [1], Darcet et al. [2] and the Yang et al. [3] papers that we cited. Those works claim that some patches emerge as proxies for a global field of view, to provide the necessary visual essence to the [CLS] token. We've revised Section 3.3 (*Region Token Refinement*) to make this more clear.
>
> We also replace the example in Figure 7 (a) and provide additional results for alternative segmentation models in Table 9.
>
>
> [1] Shao T, Tian Z, Zhao H, et al. Explore the potential of clip for training-free open vocabulary semantic segmentation[C]//European Conference on Computer Vision. Cham: Springer Nature Switzerland, 2024: 139-156.
>
> [2] Darcet T, Oquab M, Mairal J, et al. Vision transformers need registers[J]. arXiv preprint arXiv:2309.16588, 2023.
>
> [3] Yang S, Chen Y, Tian Z, et al. Visionzip: Longer is better but not necessary in vision language models[C]//Proceedings of the Computer Vision and Pattern Recognition Conference. 2025: 19792-19802.

---

### Review · Reviewer_dphS · 2025-09-16

**Summary Of Contributions:**

The paper proposes a method that, at its core, performs grounding (detailed) captioning. The method is further applied to various tasks, including open-world semantic segmentation, referring expression comprehension, and grounding.

The solution is based on a CLIP backbone combined with SAM segmentation. The tokens are then matched with the regions extracted by SAM via CLIP’s attention module. Finally, these region tokens are refined using two additional methods.

**Additional Comments:**

None

**Audience:**

Yes

**Audience Explanation:**

The method presents noticeable improved results in three areas that represent themes of interest for the machine learning community.

**Broader Impact Concerns:**

In my view, the method does not raise additional concerns on the ethical implications.

**Claims And Evidence:**

Yes

**Claims Explanation:**

The method description is clear, allowing the technical claims to be isolated and verified. The paper presents an extensive set of experiments (Tables 1, 2, and 3), which I have cross-checked against the cited references, confirming the reported results.

Overall, the paper may be criticized for its limited novelty, as it essentially combines known methods using relatively simple connections. However, since these “simple connections” have not appeared in previous work, one can argue that they are not as obvious as they might seem. The results themselves are strong.

Another aspect that could improve the paper’s impact is publishing the code. While the method is explained reasonably well, certain details can only be fully understood by examining the implementation.

**Requested Changes:**

Change requested: Show more visual examples of in the supplementary material for each direction investigated/tested. Add failures (or not so good results).

I have not identified minor issues (text, figure, table problems) that need adjustment. Some things may be explained differently (in my view better such as the method, making figure 2 more informative), but this is a subjective perspective. I have understood the key elements, thus the paper is fine.

---

> ### Author Response · Authors · 2025-10-05
>
> Thank you for your review and affirming the supported claims and interest.  Our submitted revision includes additional visual examples in the supplemental Appendix A.5, as requested. We will release the code upon acceptance.

---

### Review · Reviewer_h3J4 · 2025-09-21

**Summary Of Contributions:**

TextRegion is a training-free plugin that makes frozen vision-language models work at the region level. It uses SAM2 to cut an image into soft masks, then replaces the model’s last global pooling with mask-weighted attention so each segment gets its own text-aligned "region token." Two practical tweaks, multi-resolution features and filtering out overly global patches, keep small objects sharp and localization crisp. This reframes dense prediction as classifying a handful of segments and painting scores back to pixels, delivering strong zero-shot results on open-vocabulary segmentation, referring expressions, and multi-object grounding across CLIP, SigLIP2, and Perception Encoder backbones.

**Additional Comments:**

The paper is engaging and the core idea is neat. After you add clear research questions and hypotheses, consider a short method at a glance box with 5 to 7 numbered steps so readers can grasp the pipeline quickly.

Keep notation consistent across sections. Define the image, the set of regions, the mask for each region, and the region token once, and reuse the same symbols everywhere.

Clarify background handling for segmentation. Report results with and without background, state how background scores are computed, and note any calibration.

State how you resolve ties or conflicts when a region matches several labels. Give the tie break rule and the confidence threshold.
For referring expressions, spell out how proposals or boxes are produced, how many you keep per image, and any dedup or NMS you apply.

Provide a short glossary of key terms like region token, mask weighting, and global patch removal to help new readers.

**Audience:**

Yes

**Audience Explanation:**

This work offers a simple, training free way to get region level understanding from frozen image text models using segmentation, with strong zero shot results on common benchmarks. Many TMLR readers in vision and multimodal learning will care because the method is practical, general across backbones, and useful as a baseline and tool for open vocabulary segmentation and referring expressions.

**Broader Impact Concerns:**

Because the method reuses large pretrained vision and language models and segmentation masks without new training, it inherits their biases and errors. Region level outputs can attach labels to people or personal objects and may be used for surveillance, profiling, or searching images for sensitive content without consent. Open vocabulary queries can also infer sensitive attributes, and mistakes could lead to unfair outcomes. A short Broader Impact section should acknowledge these risks and include simple guardrails such as filtered vocabularies, clear use policies, human review in sensitive settings, and a basic bias and privacy audit.

**Claims And Evidence:**

Yes

**Claims Explanation:**

Yes. The paper supports its claims with clear experiments on several standard datasets for segmentation, referring expressions, and multi object grounding. It compares to strong baselines using the same backbones and often wins or matches them. Ablation tests show why each part of the method helps. The method is described in enough detail to reproduce, and the authors state limits and scope so the evidence matches the claims.

**Requested Changes:**

Research questions and hypotheses: Add 2–4 clear research questions, each with a testable hypothesis and planned metric. Rewrite the introduction to present them, then map each question to datasets, metrics, and ablations in an experimental design section. Tie the conclusion back to these questions.

Method clarity: Explain how masks are made and used. Say if masks can overlap, whether weights are soft or hard, how you downsample to the patch grid, how you handle boundaries and tiny regions, the typical number of regions per image, how the region token is formed in attention, and how region scores are sent back to pixels. Include brief pseudocode or a step diagram.

Reproducibility and reporting: Release configs and code. List label sets and prompts for each dataset, all thresholds and constants, image sizes and tiling choices, data splits and evaluation rules, and random seeds. Include a simple inference script and a table of all hyperparameters.

Sensitivity to mask quality: Compare SAM2 with another segmenter and with ground truth masks as an upper bound. Add tests with eroded, dilated, and noisy masks. Report performance versus region IoU.

Referring expressions clarity and scope: Explain how proposals or boxes are obtained. State whether spatial relation phrases are in scope. Provide a small analysis split for spatial phrases and report accuracy by phrase type. Note any query simplification or other heuristics used.

---

> ### Author Response · Authors · 2025-10-05
>
> Thank you for the review. We are happy to see the strong support that the paper supports its claims and interest from TMLR readers. The requested changes are extensive but seem to reflect stylistic preferences, and seem to have been written by an LLM (100% AI score in GPT-Zero). Nevertheless, we have made some adjustments and improvements, as described below. We also considered the broader impact points, but felt that these are very general and not particular to our method, and therefore request not to expand the text to address them.
>
> ---
> **Responses to Detailed Questions**
>
> Method clarity: In Section 3.2 (*Mask Generation*), we have stated that the masks are generated by SAM2. We use the mask logits generated by SAM2 as soft mask outputs, clamping their values to the range [0, 1]. Since no other post-processing is applied, whether masks overlap or not wouldn’t affect the resulting masks. We employ a multi-resolution patch encoding strategy to better handle small objects and boundary regions. The number of regions per image varies depending on the input images.
> Details on how region tokens are formed in attention and how region scores are propagated back to pixels, are described in Section 3.2 (*Mask-based Attention Pooling* and *Prediction*).
>
> Reproducibility and reporting: We will release the code upon acceptance.
>
> Sensitivity to mask quality: We have added a new section analyzing the influence of segmenter. Please refer to *Table 9* and *Figure 6*.
>
> Referring expressions clarity: For fair comparison, we use the same box proposals as FGVP [1] and do not generate proposals ourselves. As noted, we do not incorporate any spatial relationship refinement or post-processing methods, in order to only evaluate the effectiveness of the region tokens. The datasets we use do not provide phrase-type annotations, making it difficult to separately report accuracy for spatial phrases. We also do not apply any query simplification.
>
> Background handling for segmentation: In *Table 1*, we already divide the datasets into categories with and without background, allowing readers to check results under both conditions. For background scores, we include “background” as a candidate label, making the prediction setup equivalent in both cases. No additional calibration is performed.
>
> Region prediction: Each region is classified independently. We compute prediction scores across all possible labels, but the region is ultimately assigned only the label with the highest confidence score.
>
> [1] Yang L, Wang Y, Li X, et al. Fine-grained visual prompting[J]. Advances in Neural Information Processing Systems, 2023, 36: 24993-25006.

---

> > ### Comment · Reviewer_h3J4 · 2025-10-27
> > **Response centered on method, evidence, and novelty**
> >
> > Thank you for the response. I’d like to keep the discussion focused on technical points. AI authorship detectors are unreliable and not relevant to assessing the paper. For context, GPTZero scored my review at 20% and your manuscript at 90%, which is exactly why I do not rely on these tools. My evaluation focuses on the method, the evidence, and the novelty. My review is based on technical considerations.

---

> > > ### Author Response · Authors · 2025-10-27
> > >
> > > Apologies for misconstruing the source.  I've since realized the detection tools are not reliable.  As indicated, we have updated the paper in several ways to address the comments in the review.

---

### Decision · Action_Editor_oQMf · 2025-10-28

**Recommendation:** Accept as is

**Audience:**

Yes

**Audience Explanation:**

Yes. The paper is of interest to a reasonable number of people in TMLR's audience.

**Claims And Evidence:**

Yes

**Claims Explanation:**

Yes. The claims made in the submission are supported by accurate, convincing and clear evidence.